# Mapping the microscale origins of magnetic resonance image contrast with subcellular diamond magnetometry

Hunter C. Davis [1], Pradeep Ramesh[2], Aadyot Bhatnagar[3], Audrey Lee-Gosselin[1], John F. Barry[4,5,6,7], David R. Glenn[4,5,6], Ronald L. Walsworth[4,5,6] & Mikhail G. Shapiro [1]

Magnetic resonance imaging (MRI) is a widely used biomedical imaging modality that derives much of its contrast from microscale magnetic field patterns in tissues. However, the connection between these patterns and the appearance of macroscale MR images has not been the subject of direct experimental study due to a lack of methods to map microscopic fields in biological samples. Here, we optically probe magnetic fields in mammalian cells and tissues with submicron resolution and nanotesla sensitivity using nitrogen-vacancy diamond magnetometry, and combine these measurements with simulations of nuclear spin precession to predict the corresponding MRI contrast. We demonstrate the utility of this technology in an in vitro model of macrophage iron uptake and histological samples from a mouse model of hepatic iron overload. In addition, we follow magnetic particle endocytosis in live cells. This approach bridges a fundamental gap between an MRI voxel and its microscopic constituents.

[1] Division of Chemistry and Chemical Engineering, California Institute of Technology, Pasadena, CA 91125, USA. [2] Division of Biology and Biological Engineering, California Institute of Technology, Pasadena, CA 91125, USA. [3] Division of Engineering and Applied Sciences, California Institute of Technology, Pasadena, CA 91125, USA. [4] Harvard-Smithsonian Center for Astrophysics, Harvard University, Cambridge, MA 02138, USA. [5] Department of Physics, Harvard University, Cambridge, MA 02138, USA. [6] Center for Brain Science, Harvard University, Cambridge, MA 02138, USA. [7] Present address: MIT Lincoln Laboratory, Lexington, MA 02420, USA. Hunter C. Davis and Pradeep Ramesh contributed equally to this work. Correspondence and requests for materials should be addressed to M.G.S. (email: mikhail@caltech.edu)

Magnetic resonance imaging (MRI) is a widely used biomedical imaging modality, with millions of scans performed each year for medical diagnosis, human neuroscience research, and studies in animal models. The contrast seen in MRI images is strongly influenced by microscale magnetic field gradients in cells and tissues, produced by endogenous substances such as blood, cellular iron deposits[1,2], or molecular-imaging agents such as iron oxide nanoparticles (IONs)[3–6]. The precise dependence of voxel-scale (~0.5 mm) MRI contrast on the microscale magnetic field has been a topic of intense theory and simulation due to its importance for disease diagnosis and contrast agent design[2,7–10]. These studies predict, for example, that the spatial frequency of the local magnetic field can significantly impact the $T_2$ relaxation rate of a tissue, and that

optimizing contrast agent size can maximize $T_2$ contrast for a given set of material and imaging parameters. However, despite its significance for biological imaging, the relationship between microscopic magnetic field patterns in tissue and $T_2$ relaxation has not been studied experimentally due to a lack of effective methods to map magnetic fields at the microscale under biologically relevant conditions.

Nitrogen-vacancy (NV) magnetometry is a recently developed technique that enables the imaging of magnetic fields with optical resolution using the electronic properties of fluorescent NV quantum defects in diamond[11]. The electronic structure of an NV center forms a ground-state triplet, with the $m_s = \pm 1$ states separated from the $m_s = 0$ state by 2.87 GHz, making ground-state spin transitions addressable by standard electron spin resonance

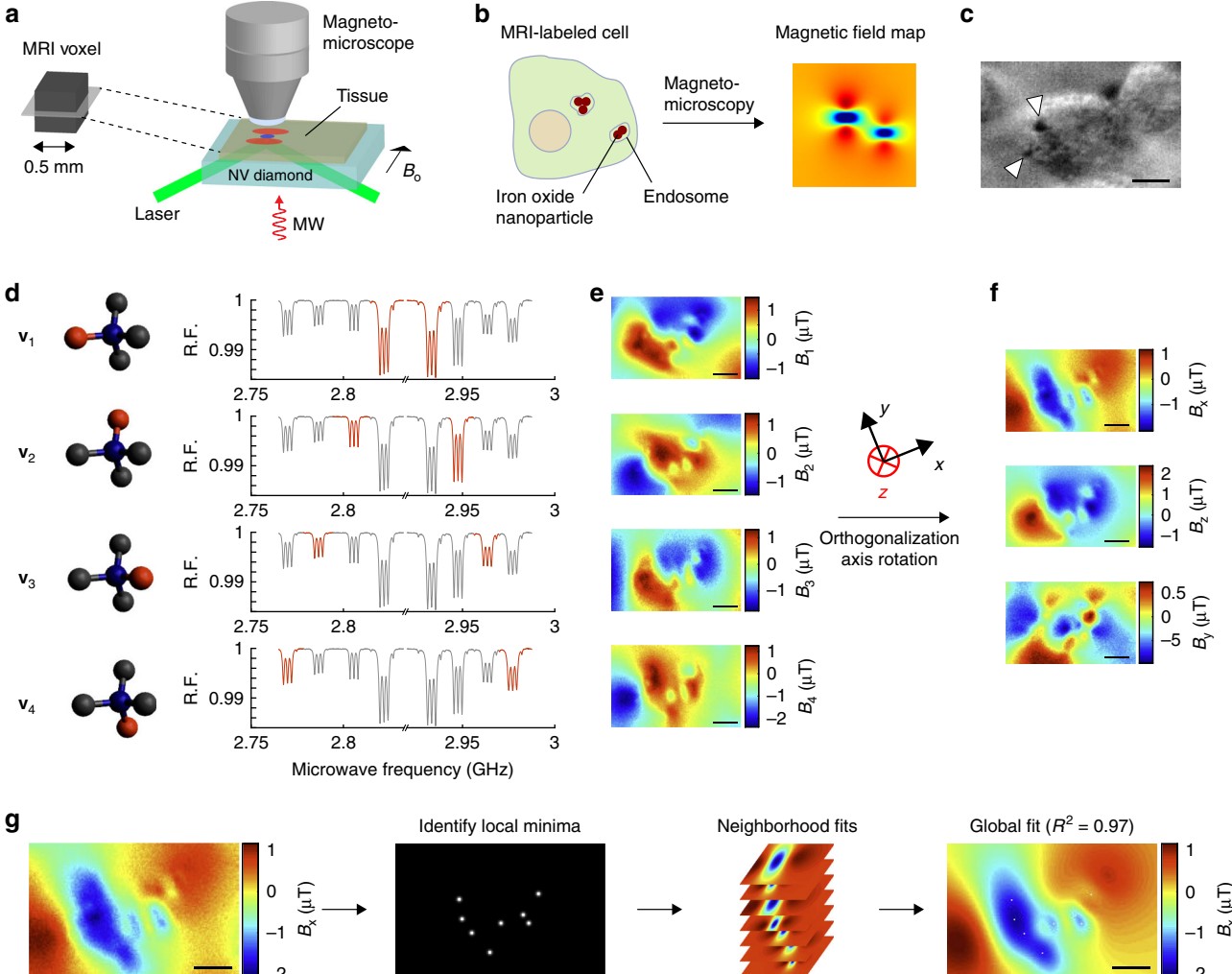

**Fig. 1** Subcellular mapping of magnetic fields in cells labeled for MRI. **a** Schematic of subvoxel magnetic field mapping using a NV magneto-microscope. **b** Illustration of a cell labeled with IONs and its expected magnetic field pattern. **c** Bright-field image of RAW 264.7 macrophage labeled with 200-nm IONs. White arrows point to internalized IONs. A bright-field imaging artifact also appears as black in the upper right corner of the cell. **d** Cartoon representation of each NV orientation and the corresponding representative spectra from fixed-cell experiments. The blue ball represents nitrogen and the red ball represents the adjacent lattice vacancy. Highlighted peaks in each relative fluorescence (RF) spectrum show the transition corresponding to each of the four orientations. **e** Magnetic field images of the field projections along each of the four NV axes of macrophages 2 h after initial exposure to 279 ng ml$^{-1}$ 200-nm IONs. **f** Images in **e** converted via Gram–Schmidt orthogonalization and tensor rotation to field maps along three Cartesian coordinates with the $z$ axis defined perpendicular to the diamond surface and the $x$ axis defined as the projection of the applied bias field onto the diamond surface plane. The $y$ axis is defined to complete the orthogonal basis set. **g** Representative example of the procedure for dipole localization in cellular specimens. This procedure comprises three steps: first the local minima in the field map are identified and ranked; next, in decreasing order of magnitude, the neighborhood of each local minimum is fit to a point dipole equation and the resulting field is subtracted from the field map to reduce the fit-deleterious effect of overlapping dipole fields; and finally, the results of these fits are used as guess parameters for a global fit over the full field of view. The fit shown has a degree-of-freedom-adjusted $R^2$ of 0.97. Scale bars are 5 μm

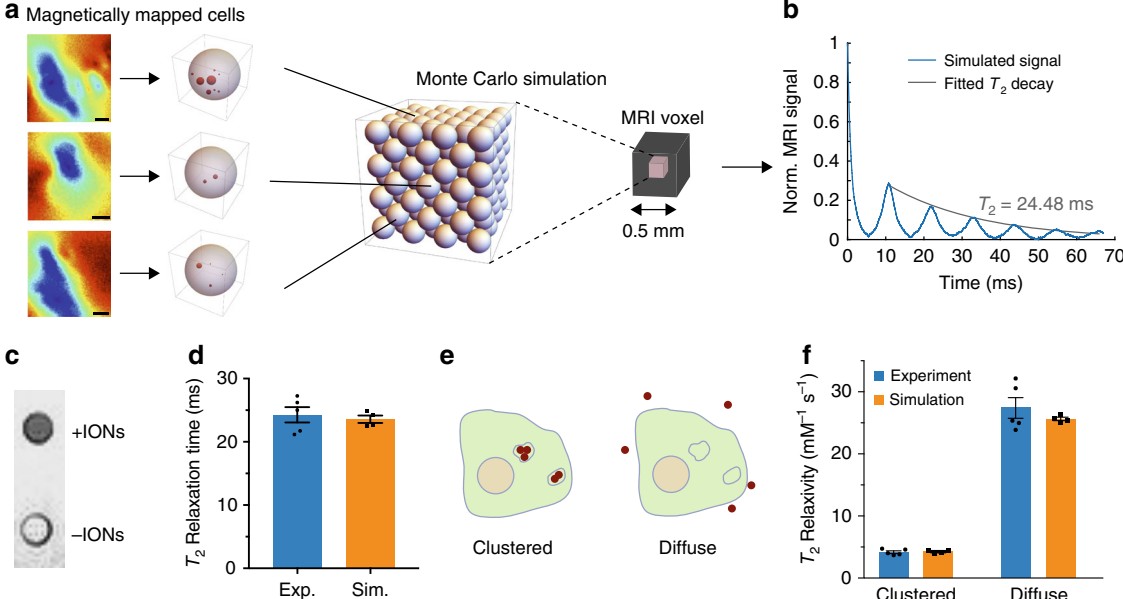

**Fig. 2** Predicted and experimental MRI behavior in cells. **a** Schematic of Monte Carlo modeling of spin relaxation using NV-mapped magnetic fields. A library of 11 cells mapped with vector magnetometry (three representative cells shown) in a 1:1 mix with unlabeled cells, was used to randomly fill a 108-cell FCC lattice with periodic boundary conditions and run a Monte Carlo simulation of spin-echo MRI to predict $T_2$ relaxation behavior. **b** Representative simulated MRI signal. **c** $T_2$-weighted MRI image of cell pellets containing a 1:1 mixture of supplemented and unsupplemented cells (+ IONs and –IONs, respectively) or 100% unlabeled cells (bottom). **d** Simulated and experimentally measured $T_2$ relaxation times for the 1:1 mixture. **e** Illustration of the same quantity of magnetic particles endocytosed or distributed in the extracellular space. **f** Simulated and experimentally measured relaxivity for endocytosed and extracellular distributions of IONs. Measurements and simulations have $N = 5$ replicates. All error bars represent ± SEM

(ESR) techniques. The Zeeman energy difference between the + 1 and −1 states leads to the splitting of the 2.87 -GHz resonance into two distinct energy levels, whose separation from each other increases linearly with magnetic field strength. Upon green laser excitation (532 nm), the $m_s = \pm 1$ states are more likely to undergo non-radiative relaxation than the zero-spin state, so that microwave-induced transitions from $m_s = 0$ to $m_s = \pm 1$ cause a drop in NV fluorescence. Thus, the local magnetic field of an NV center can be extracted from the optically reported ground-state spin transition frequency. Diamonds densely doped with NV centers make it possible to optically image this resonant transition frequency over a wide field of view, thus providing an Abbe-limited image of the magnetic field at the diamond surface[12]. NV magnetometry has recently been used in proof-of-concept biological applications such as imaging the magnetic fields produced by magnetotactic bacteria[13], detecting magnetically labeled cancer cells[14], visualizing paramagnetic ions bound to cells[15], and measuring magnetic fields produced by neuronal action potentials[16].

Here, we establish a method that uses the unique capabilities of NV magnetometry to study the connection between subcellular magnetic fields and MRI contrast. Doing so requires adapting NV magnetometry for high-sensitivity imaging of sparse magnetic fields in cells and tissues, developing methods to convert two-dimensional (2D) NV data into the three-dimensional (3D) distribution of magnetic field sources, and simulating the behavior of nuclear spins in the resulting magnetic fields. In addition, monitoring the evolution of magnetic fields in live cells requires operating under nondamaging optical and thermal conditions with reduced available signal. In this work, we address these challenges to enable the mapping of subcellular magnetic fields in an in vitro model of macrophage iron oxide endocytosis and histological samples from a mouse model of liver iron overload, connecting both to MRI contrast.

## Results

**Mapping subcellular magnetic fields**. Our home-built NV magneto-microscope (Fig. 1a) was optimized for both high-resolution magnetic field imaging of fixed samples and dynamic imaging of living cells. By virtue of a relatively thick NV layer in our diamond (~4 μm), we were able to significantly reduce the applied laser power compared to shallower surface-implanted NV diamond microscopes, while maintaining a strong NV fluorescent signal for rapid imaging. We used a total internal reflection geometry to minimize phototoxicity[13,16] and bonded a silicon carbide wafer to the diamond base to improve thermal dissipation[16]. For cell-imaging experiments, we applied a moderate bias field (10 mT) to magnetize cell-internalized superparamagnetic IONs. While a larger bias field would increase the magnetization of the sample, it would also produce stronger off-axis magnetic fields for each NV axis, which significantly reduces the sensitivity of NV magnetometry[17].

As a first test of our method, we imaged the magnetic fields resulting from the endocytosis of superparamagnetic IONs by murine RAW 264.7 macrophages. Magnetic labeling and in vivo imaging of macrophages are under development for a variety of diagnostic and therapeutic applications[4,18–20], which could benefit from an improved understanding of the resulting MRI contrast. In particular, although labeling is typically done with dispersed particles of sizes ranging from a few nanometers to several microns[21–23], their internalization and subsequent compaction by the cell (Fig. 1b, c) could produce radically different magnetic field profiles[8–10], which cannot be directly observed by conventional electron microscopy or iron-staining techniques. We performed vector magnetometry on fixed macrophages after incubating them for 1 h with 200 -nm, multicore IONs and allowing one additional hour for internalization. After measuring the magnetic field along each of the four NV orientations (Fig. 1d), we projected the field maps along

Cartesian axes convenient for magnetic dipole localization via orthogonalization and tensor rotation (Fig. 1e, f).

**Connecting microscale fields to MRI contrast**. To connect microscale magnetic field measurements to MRI contrast, we first converted our 2D images to 3D maps of magnetic field sources in the sample, and then simulated the behavior of aqueous nuclear spins in the corresponding 3D field. To convert 2D vector maps imaged at the diamond surface into a 3D model of magnetic fields in cells above the diamond, we developed an algorithm for iterative localization of magnetic dipoles (Fig. 1g, Supplementary Fig. 1). First, the in-plane coordinates of putative dipole field sources (clusters of magnetic particles) were identified from local minima in the $x$ component of the vector field, chosen parallel to the projection of the bias field onto the diamond surface. Then, the off-diamond height ($z$) and magnetic moment of each cluster were determined by fitting the local dipole field profile. After fitting the dipole at the strongest local minimum, the resulting magnetic field pattern was subtracted, and the next strongest local minimum fitted, with this process repeated until all local minima were exhausted. A global fit was then performed using the results from the local fits as starting parameters. The degree-of-freedom-adjusted $R^2$ for all the global fits made to six representative particle-containing cells was greater than 0.90. Magnetic localization of nanoparticle clusters was confirmed in a separate set of cells using fluorescently labeled nanoparticles (Supplementary Fig. 2). In addition, independent measurements of intracellular iron concentration using inductively coupled plasma mass spectroscopy, $1.09 \pm 0.10$ pg Fe per cell, corroborated the estimated iron content inferred from NV measurements, which was 1.126 pg Fe per cell. The final dipole values were scaled from the 10-mT bias field of the NV instrument to the 7-T field of our MRI scanner using the bulk magnetization curve of the IONs (Supplementary Fig. 3, Supplementary Note 1).

To translate subcellular magnetic field maps into predictions about MRI contrast, we performed Monte Carlo simulations of nuclear spin $T_2$ decoherence in lattices of representative cells. These cells contained magnetic dipole distributions and magnitudes derived from NV magnetometry of a representative cellular library (Fig. 2a, Supplementary Fig. 4). The resulting lattice thereby contains information about the spatial frequencies of the magnetic field present in the pellet tissue, a critical parameter for $T_2$ contrast. Importantly, since this information can be obtained from NV measurements performed on a representative sampling of cells or tissues, this obviates the need for NV evaluation of the exact individual sample imaged with MRI, enhancing the versatility of this approach.

Our simulation predicted a bulk MRI $T_2$ relaxation time of 23.6 ms for a 1:1 mixture of supplemented and unsupplemented cells (Fig. 2b). Mixing was done to obtain a sufficiently long $T_2$ for accurate measurement with our MRI system. When compared to an experimental MRI measurement of $T_2$ in macrophages prepared as in the NV experiment and pelleted in a 1:1 mixture with unsupplemented cells, the Monte Carlo prediction was accurate to within 2.8% (Fig. 2c, d). The $T_2$ relaxation time of the cell pellets could not have been predicted solely from the concentration of IONs in the sample, as previous simulations have suggested a major influence of packing geometry on contrast agent relaxivity[8–10]. To establish that this relationship also holds for our model system, we performed MRI measurements and Monte Carlo simulations with IONs distributed in the extracellular space (Fig. 2e). Per iron mass, we found that this diffuse extracellular arrangement produces approximately sixfold faster $T_2$ relaxation than do endocytosed particles (Fig. 2f), underlining the importance of the microscale magnetic field patterns mapped with our method. Simulations of additional particle distributions examine the relative influence of particle clustering and confinement inside cells and endosomes (Supplementary Fig. 5, Supplementary Note 2).

**Mapping magnetic fields in histological specimens**. To extend this technique to diagnostic imaging, we performed NV magnetometry on liver specimens from a mouse model of hepatic iron overload. The spatial distribution of iron deposits in the liver and other tissues has been a topic of interest in clinical literature as an indicator of disease state, including efforts to discern it non-invasively using MRI[2]. Iron overload was generated through intravenous administration of 900-nm IONs to C57bl/6 mice (Fig. 3a). Livers were harvested 18 h after injection and imaged with 7-T MRI, showing enhanced macroscale $T_2$ relaxation compared to controls (Fig. 3b). To investigate the microscale nature of this contrast enhancement, we cryosectioned the livers of saline- and iron-injected mice and imaged the magnetic field profiles of these tissue sections on our NV magneto-microscope. We measured the projection of the magnetic field along a single NV orientation, probing the $m_s = 0$ to $m_s = +1$ and $m_s = 0$ to $m_s = -1$ transitions. The magnetic particle clusters were relatively sparse, resulting in a punctate distribution of magnetic dipoles within the liver tissue of the iron-overloaded mouse (Fig. 3c, Supplementary Fig. 6). We confirmed that these magnetic fields

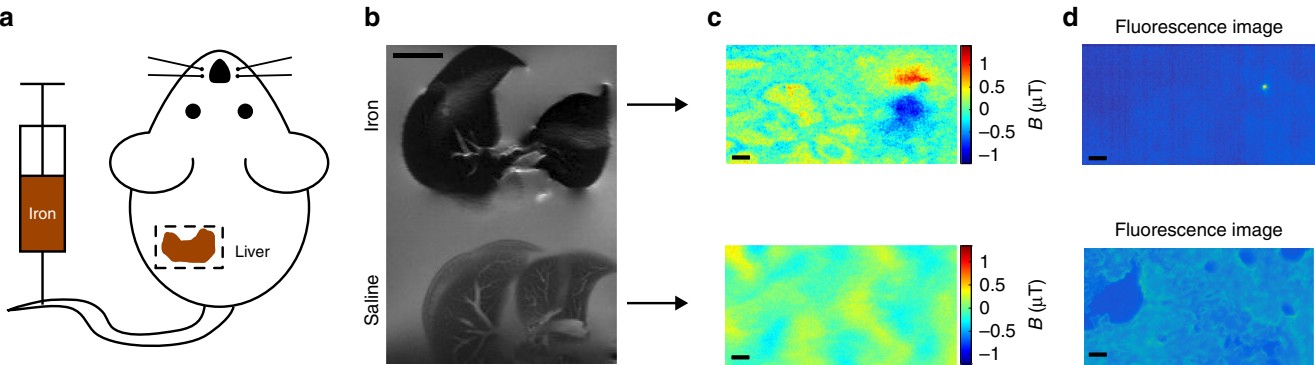

**Fig. 3** Magnetometry of histological samples. **a** Diagram of mouse model of iron overload, prepared by injecting 10 mg kg$^{-1}$ of 900 nm iron oxide nanoparticles into the tail vein. **b** 7T $T_2$-weighted MR image of fixed, excised mouse livers from mice injected with IONs or saline. **c** NV magnetic field maps of 10 μm liver sections obtained from the mice in **b**. **d** Fluorescence images of the tissue samples in **c**. Fluorescence images were taken with autogain to reduce the necessary exposure time, resulting in the visibility of the autofluorescence of the tissue in the saline control. Magnetometry scans were taken with a fixed gain. This experiment was repeated a total of three times, with data from two additional experiments shown in Supplementary Fig. 6. Scale bars in **b** and **c–d** are 5 mm and 10 μm, respectively

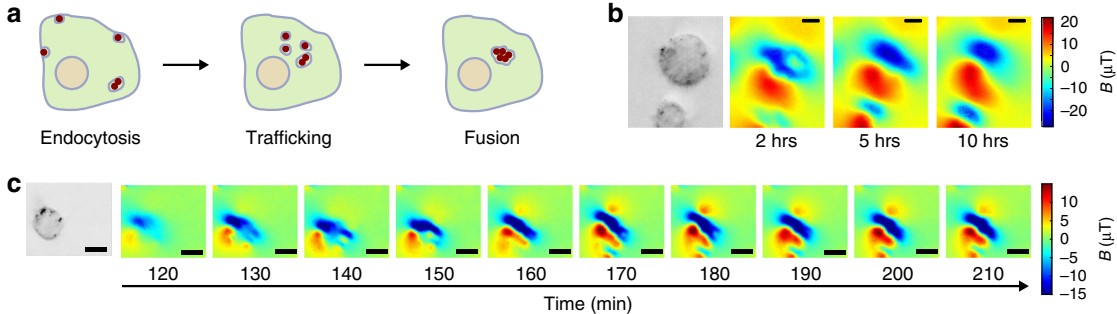

**Fig. 4** Dynamic magnetic microscopy in live mammalian cells. **a** Cartoon showing the typical progression of endocytotic uptake of IONs. **b** Bright field and series of time-lapse magnetic field images of RAW macrophages over 10 h. Three additional replicates are shown in Supplementary Fig. 7. **c** Bright field and series of time-lapse magnetic field images of a RAW macrophage with 10 min between magnetic field images. Two additional replicates of this experiment are shown in Supplementary Fig. 7. Scale bars are 5 μm

resulted from IONs using fluorescent imaging, for which purpose the IONs were labeled with a fluorescent dye (Fig. 3d). These results suggest that NV magnetometry could be used to map subvoxel magnetic field patterns within histological specimens, increasing the diagnostic power of MRI by correlating magnetic field distributions to disease state.

**Magnetic imaging of endocytosis.** Finally, we tested whether NV magnetometry could be used to follow the magnetic consequences of the dynamic redistribution of magnetic material in living mammalian cells. Macrophages endocytosing IONs go through several stages of internalization, gradually reconfiguring diffuse particles into compacted lysosomal clusters (Fig. 4a). This process could be relevant to interpreting MRI data from labeled macrophages and to the development of clustering-based magnetic nanoparticle contrast agents[24,25]. To image living cells, we adjusted our NV methodology to minimize optical and thermal energy deposition. We subsampled the NV spectrum to probe only the $m_s = 0$ to $m_s = +1$ transition of one NV orientation and limited laser illumination to 5 min per image. This allowed us to generate time-lapse images of magnetic fields coalescing inside macrophages after ION internalization (Fig. 4b, c, Supplementary Fig. 7), at the expense of precise 3D source localization, which requires vector magnetometry using multiple NV orientations. Cell viability (assessed via a Trypan Blue exclusion assay) was approximately 90%. This technique for magnetic imaging of a dynamic cellular process could aid the development of dynamic contrast agents for MRI.

## Discussion

In summary, this work establishes the capability of subcellular NV diamond magnetometry to map microscale magnetic field patterns in mammalian cells and tissues and introduces computational methods to connect these patterns to MRI contrast. The ability to make this connection experimentally will facilitate the interpretation of noninvasive images through microscopic analysis of matching histological specimens, and aid the development of magnetic contrast agents for molecular imaging and cellular tracking. Alternative methods for magnetic measurement, such as scanning superconducting quantum interference device (SQUID) microscopy[26,27] and magnetic force microscopy[28,29], are more difficult to apply to tissue-scale biological specimens due to the need to raster scan samples, the spatial offsets required for thermal insulation of SQUID magnetometers from biological materials, and the need to penetrate samples with probe tips for force microscopy. MRI itself can also be used at higher resolution to examine ex vivo specimens, but does not typically approach the single-micron level[30,31]. Meanwhile, methods such as electron

microscopy or iron staining, which can also reveal the in vitro locations of putative magnetic materials based on their density or atomic composition, contain no information about the magnetic properties of such materials and their resulting fields, limiting the utility of these methods to examining the distribution of known magnetic field sources.

Although the present study also used known particles to enable direct experimental validation of our methods, NV magnetometry can in principle be used to measure magnetic field profiles arising from unknown sources, such as biomineralized iron oxide. To enable such measurements, NV imaging could be performed with a variable, electromagnet-driven bias field to first map the locations of magnetic field sources at low field (where vector magnetometry is possible), and then apply a ramping field along a single NV axis to assess the $M$ versus $H$ behavior of each field source. Such in situ saturation curves would provide the information needed to model MRI relaxation in samples with unknown saturation behavior. Additional improvements in this technique may be needed to reconstruct the location and magnetization of more diffuse magnetic materials that are less easily detected as point dipoles.

The sensitivity of our current instrument, established by computing the variance between three sequential magnetic measurements of the identical sample, was 17 nT at 1-μm in-plane resolution. This sensitivity corresponds to the field produced by a 92-nm particle situated 10 μm above the diamond surface (assuming the same volumetric magnetization as the IONs used in this study), or a 10-nm particle located immediately on top of the diamond. This sensitivity was more than sufficient to detect the 200-nm IONs used in our proof-of-concept experiments. While these particles are within the size range used in MRI contrast agents[21–23], future work should focus on improving the sensitivity of NV magnetometry and demonstrating detection of smaller sources. Sensitivity could be improved by employing diamonds with thinner NV layers, which would allow detection of significantly smaller magnetic sources near the diamond surface and would reduce the point-spread function of NV-imaged magnetic fields, increasing the precision of source localization. Combined with improved methods for positioning tissue sections flatter on the diamond surface, this would allow the mapping of fields produced by smaller, endogenous magnetic inclusions and ultrasmall superparamagnetic nanoparticles.

The study of microscale sources of $T_2$ contrast could be complemented by methods to map the concentrations of $T_1$ contrast agents using alternating current (AC) NV magnetometry[15]. In particular, adapting this technique to measure the 3D distribution of $T_1$ agents inside of the cell using

nanodiamonds[32,33] could enable Monte Carlo modeling of $T_1$ relaxation in contrast-labeled cells and tissues. In addition to mapping the distribution of contrast agents and the resultant magnetic fields, recent advances in NV magnetometry could allow for in situ imaging of water-bound proton relaxation, enabling a direct measurement of the effect of contrast agents on the relaxation of surrounding water molecules[34].

Besides contributing to the study of MRI contrast, the methods presented for mapping magnetic field sources in 3D from planar optical data will enable biological imaging applications directly using NV diamonds and magnetic labels. Because the optical readout in this technique is confined to the diamond surface, this method can be used to study opaque tissues inaccessible to conventional microscopy. To this end, our demonstration that time-resolved wide-field NV magnetic imaging can be performed on living cells increases the utility of this technique for monitoring dynamic biological processes.

## Methods

**Nitrogen-vacancy magneto-microscope.** The NV magneto-microscope was constructed from a modified upright Olympus BXFM microscope and a 532 nm laser source. The diamond used in this work is an electronic grade ($N < 5$ p.p.b.) single crystal substrate with nominal rectangular dimensions of 4.5 mm × 4.5 mm × 500 μm, grown using chemical vapor deposition (CVD) by Element Six. The top-surface NV sensing layer is measured to be 3.87 μm thick, consists of 99.999% isotopically pure $^{12}$C with 21.4 p.p.m. $^{14}$N ($3.77 \times 10^{17}$ cm$^{-3}$) incorporated into the layer during growth. Layer thickness and nitrogen concentration were determined by secondary ion mass spectroscopy. The diamond was irradiated with a 4.5 MeV electron source with an irradiation dose of $9 \times 10^{18}$ cm$^{-2}$. The samples were subsequently annealed at 400 °C for 2 h, 800 °C for 16 h, and 1200 °C for 2 h. This diamond was affixed to a silicon carbide wafer (for enhanced heat dissipation), which was in turn affixed to a pair of triangular prisms to facilitate a total internal reflection excitation path. The prisms, silicon carbide wafer and diamond were fused using Norland Optical Adhesive (NOA 71). The diamond assembly was removable to allow live-cell culture on the diamond surface in a cell culture incubator. Light was collected from the top of the diamond through a water-immersion objective. Images were acquired on a Basler acA2040-180kmNIR—CMV4000 CCD camera with 2048 × 2040 5.5 μm pixels (we used 256 × 1020 pixels to increase frame rate). For high-resolution vector magnetometry and tissue imaging, NV fluorescence was excited using a 100 mW Coherent OBIS LS 532 nm optically pumped semiconductor laser. For live-cell imaging, we used an attenuated 2 W 532 nm laser from Changchun New Industries Optoelectronics. When necessary, focal drift was adjusted for using a piezo-driven stage (Thorlabs). Microwave radiation was applied through a single turn copper loop immediately surrounding the diamond. The microwave signal was generated by a Stanford Research Systems Inc. SG384 signal generator and amplified by a ZHL-16W-43-S + amplifier from MiniCircuits. Experimental timing was controlled by a National Instruments USB 6363 X Series DAQ. A bias magnetic field was generated by two NeFeB grade N52 magnets (1″ × 2″ × 0.5″, K&J Magnetics) positioned on opposite sides of the NV diamond. The NV setup was controlled by custom software written in LabView.

**Cell culture.** RAW 264.7 cells (ATCC) were cultured at 37 °C and 5% CO$_2$ in Dulbecco's Modified Eagle Medium (DMEM, Corning Cellgro) and passaged at or before 70% confluence. For particle labeling, media was aspirated and replaced with phenol red-free DMEM supplemented with 279 ng ml$^{-1}$ IONs (200 nm Super Mag Amine Beads Ocean Nanotech, MHA). After 1 h, the ION solution was aspirated and cells were washed twice with phosphate buffered saline (PBS) to remove unbound particles. For fixed-cell magnetometry, the cells were trypsinized, quenched with DMEM and deposited on the diamond surface at 40–70% confluency. After 1 h incubation on the diamond under ambient conditions, the cells were fixed with 4% paraformaldehyde-zinc fixative (Electron Microscopy Services) and washed twice with PBS.

For live-cell imaging, the cells were cultured as above until trypsinization and spotting on the diamond. Their media was supplemented with 0.1 mM ascorbic acid to mitigate phototoxicity[35]. For extended imaging, the cells were maintained on the diamond in DMEM supplemented with 10 mM HEPES to stabilize pH at 7.4 under ambient atmosphere.

**Vector magnetometry.** The bias magnetic field was aligned close to in-plane with the diamond surface while having sufficient out-of-plane field strength to resolve the resonance of each NV axis, and the full NV optically detected magnetic resonance (ODMR) spectrum was probed. The out-of-plane component was necessary because a purely in-plane bias field did not provide each NV axis with a unique parallel $B$-field, causing absorption lines to overlap. The microwave

resonance for each pixel in the image was set as the center of the middle hyperfine peak of the transition. Spectra were swept at 0.5 Hz with 2000 images acquired per spectrum (0.9 ms exposure time). Images were acquired with an Olympus 60× water immersion objective (NA 1.0). Magnetometry spectra were acquired for 2 h each. For a sub-set of measurements, this time was extended to 6 h to improve signal-to-noise ratio (SNR).

Projection field maps for each NV orientation were generated from the corresponding peaks in the NV ODMR spectrum, and the background magnetic gradient from the bias magnets (32 μT mm$^{-1}$ in a representative scan) was subtracted out by fitting the background to a 2D quadratic function and subtracting the fit from the signal. Projection field maps were combined to form 3 orthogonal field maps with $B_z$ oriented normal to the diamond sensing surface. $B_x$ is defined as the projection of the applied field onto the diamond plane and $B_y$ is defined along the vector that completes the orthogonal set. Pixels were binned 2 × 2 in post-processing to boost SNR. This does not cause a significant reduction in resolution, as the binned pixels in the object plane are 92 nm on a side, which oversamples the Abbe limit of ~340 nm.

**Live cell magnetometry.** For live cells, the bias magnetic field was aligned such that it was possible to resolve at least one NV resonance, and the magnetic field projection along a single NV orientation was probed using the $m_s = 0 \rightarrow m_s = +1$ transition. The microwave resonance for each pixel in the image was set as the center of the middle hyperfine peak of the transition. While probing only one NV transition allowed us to reduce the light dose to the sample while maintaining good SNR, it also limited our information to a projection of the field along one axis. This limitation precludes the source fitting performed on the fixed samples. Spectra were swept 10 MHz at 1 Hz with 200 images acquired per spectrum (4 ms exposure time). In order to limit phototoxicity, each image was averaged for only 5 min and the laser was shuttered for 5 min in between images, resulting in a 50% duty cycle. Regions of interest were selected to include all relevant fields for a given cell. Optical power density was ~40 W cm$^{-2}$. Images were acquired with a Zeiss 40× near infrared water immersion objective (NA 0.8). Cell viability was assessed by performing a Trypan Blue exclusion assay after NV measurements.

**Intracellular iron quantification.** We performed inductively coupled plasma mass spectrometry (ICP-MS) to independently confirm the intracellular iron concentration estimated by NV magnetometry. RAW 264.7 cells were cultured and labeled with IONs as described above. After trypsinizing, the cells were counted using a disposable hemocytometer (InCYTO C-Chip). The cells were then pelleted at 400 g for 5 min, and the supernatant was aspirated. The cell pellet was first boiled in 2 mL of 70% nitric acid (ICP grade, Sigma) for 24 h to completely oxidize and dissolve any intracellular iron. The dried residue was then resuspended in 2% nitric acid and diluted 10-fold with deionized water for analysis using an Agilent ICP-MS quadrupole spectrometer. Unsupplemented cells contained 0.21 +/− 0.04 pg Fe per cell. A procedural blank was included throughout the process to account for background iron contamination (~34 p.p.b.), which was subtracted from measured samples.

**Field fitting and dipole localization.** In-plane dipole coordinates were identified as local minima in the $B_x$ field map. Before localization, the field map was spatially low-passed (2D Gaussian filter with $\sigma = 0.5$ pixels) to eliminate noise-generated local minima in the background. A pixel was identified as a local minimum if and only if its $B_x$ field value was smaller than all of its immediate neighbors (including diagonals) in the spatially low-passed image.

Starting with the strongest local minimum, the measured magnetic field in a 10 × 10 pixel (1.8 × 1.8 μm) square surrounding this minimum was fitted to a point dipole equation and averaged through the full NV layer depth (assuming uniform NV density), with the magnetic moment, height off of the diamond, and dipole orientation as free parameters.

$$B_x(i,j) = \frac{\int_{-z}^{-(z+h)} B_{xo}(i', j', b, \mathbf{M}, \theta, \phi) \cdot \mathrm{d}b}{-h}$$

where

$$B_{xo}(i,j) = \frac{\mu_0}{4\pi} \cdot \left( \frac{3x(\mathbf{M} \cdot \mathbf{r})}{r^5} - \frac{\mathbf{M} \cdot \hat{x}}{r^3} \right)$$

Here $i' = (i - i_0)$ and $j' = (j - j_0)$, where $(i_0, j_0)$ are the in-plane coordinates of the magnetic dipole, $\theta$ and $\phi$ correspond to the in-plane and out-of-plane angles, respectively, of the point dipole orientation, $\mathbf{M}$ is the magnetic moment, $z$ is the height of the dipole over the diamond, $\mathbf{r}$ is the displacement vector, $\hat{x}$ is the unit vector along the projection of the dipole axis onto the diamond surface plane, $x = i' \cos(\theta) - j' \sin(\theta)$, $b$ is a dummy variable for integration through the NV layer, and $h$ is the NV layer thickness. All parameters are free to fit other than the in-plane dipole coordinates, which are fixed by the local minimum of the $B_x$ field map. While the $z$ offset between the dipole and the diamond and the magnetic moment of the dipole both affect the strength of the detected field, they have distinguishable effects on the resultant field pattern. This is clear from the distinct dependence of the dipole function on $\mathbf{M}$ and $z$ (Supplementary Note 3).

After the strongest minimum has been fitted, the fitted field from the fit dipole (within the full field of view) was subtracted from the magnetic field image, to facilitate the fitting of weaker dipoles. The $10 \times 10$ pixel neighborhood of the second strongest dipole was then fitted in the subtracted image. The fitted field was subtracted, and the fitting continued until the list of local minima had been exhausted.

A global fit was then performed using the results from the neighborhood fits as starting parameters. The global fit function is the sum of $N$ dipoles (where $N$ is the number of local minima) with the in-plane dipole coordinates fixed at the local minima.

$$B_{x_{tot}}(i,j) = \sum_q B_{x_q}(i,j)$$

Here $q$ is an index that runs from one to $N$ and indicates the dipole field source. The precision of this technique is limited by the key assumption that the local minima are not significantly shifted in the $x$–$y$ plane by neighboring dipoles. The degree of freedom-adjusted $R^2$ for each of the four global fits in the cell library was greater than 0.9. For 3 of the 6 labeled cells, with image acquisition time increased from 2 to 6 h, the $R^2$ was greater than 0.95. While this approach was able to produce a sufficiently precise magnetic field reconstruction to predict MRI relaxation, other methods are also available for analytic dipole localization and magnetic field reconstruction[36].

**Fluorescent colocalization**. For fluorescence colocalization, IONs were labeled at their amino groups with Alexa 488-NHS (ThermoFisher Scientific). Before labeling, nanoparticles were diluted to 1 mg ml$^{-1}$ in 0.1 M sodium bicarbonate at pH = 8.2. Alexa 488 dye was dissolved in dimethyl sulfoxide (DMSO) at 10 mg ml$^{-1}$ and added in 10 times molar excess to the nanoparticle surface amino groups. Fluorescent images were taken before the NV magnetometry commenced to avoid photobleaching due to NV illumination. A 2-h vector magnetometry scan was then performed for localization of magnetic field sources. Alexa 488 fluorescent signal was Wiener filtered to remove background speckle and then Gaussian blurred. Local maxima of the Gaussian blurred image were designated the centroids of the fluorescent signal. In one case, we were unable to establish a fluorescent centroid corresponding to a dipole that was visible on the NV magnetometry scan. Fitting of this magnetic source predicted a magnetic moment corresponding to a single nanoparticle, which may explain its weak fluorescent signal.

**Monte carlo simulations and cell library**. Nuclear spin relaxation was simulated by assigning 11 representative cells from vector magnetometry to random positions in a repeating face-centered cubic (FCC) lattice containing a total of 108 spherical cells with periodic boundary conditions. The intracellular volume fraction of this packing geometry is 74%. While spherical cells in a periodic lattice represent a geometric simplification compared to real tissues, this and similar simplifications have been used previously to model diffusion in cell pellets and tissues[37–39]. Cell size was set to match previously measured values for RAW 264.7 cells[40]. Water molecules were randomly assigned initial $x$, $y$, and $z$ coordinates in the lattice and allowed to diffuse while their phase in the rotating frame evolved from $\phi(0) = 0$ by $\delta\phi(t) = -\gamma B_x(x,y,z)\delta t$, where $B_x(x,y,z)$ is the component of the local nanoparticle-induced field along the MRI bias field. This phase step does not account for inner-sphere effects from water coordinating to the nanoparticle surface, which will cause rapid dephasing of water coordinated to the ION surface that cannot be refocused by the pi pulses in the CPMG sequence. Re-focusing pulses were simulated at 5.5 ms Carr–Purcell time (11 ms echo time) by setting $\phi(t) = -\phi(t - \delta t)$ Adjusting the Carr–Purcell time can affect the determination of $T_2$. We used an 11 ms echo time to match the echo time of our cell pellet MR measurements. The magnetic field was mapped within this 3D-volume using a finite mesh whose mesh size was inversely proportional to the local field gradient. If a water molecule moved within a distance equivalent to six nanoparticle cluster radii of a cluster, the field contribution from that cluster was calculated explicitly. Background RAW cell relaxation was accounted for by post-multiplying the simulated signal with an exponential decay with time constant set to the measured relaxation rate of an unlabeled RAW cell pellet. Cell membranes were modeled as semi-permeable boundaries with a permeability of .01 μm ms$^{-1}$ in accordance with previously measured values for murine macrophage-like cells, adjusted to the temperature in our magnet bore (12.9 °C)[41]. Intracellular and extracellular water diffusivity were set, respectively, to 0.5547 and 1.6642 μm$^2$ ms$^{-1}$ in accordance with previous studies of cellular diffusion[37,38,42] and established values for water diffusivity at 12.9 °C[43], the temperature of our scanner bore. Bulk spin magnetization in the sample was calculated as $M(t) = \sum_i \cos[\phi_i(t)]$, where $i$ is the index of simulated water molecules and the magnetic moment of a single molecule is normalized to 1.

Nanoparticle clusters were modeled as spheres packed so as to occupy three times the volume of their constituent nanoparticles, within the range of measured literature values and grain packing theory[44–46]. To account for the increase in nanoparticle magnetizations at 7 T compared to our NV bias field, we scaled dipole magnetization using a SQUID-measured curve (Supplementary Fig. 3). Magnetic dipole coupling effects between particles were neglected, as is valid for our average cluster size and geometry. (See Supplementary Information for further discussion). The data presented in the manuscript represents the output of $N = 10$ simulations,

each containing 20 random arrangements of cells and 2000 water molecules. The number of trials was chosen such that the SEM for our simulations was smaller than the SEM of our corresponding experiments.

To assess the impact of an alternative nanoparticle distribution (Fig. 2e, f; Supplementary Fig. 5), we simulated the same 200 nm nanoparticles in the arrangements indicated in the figures. The presented data comprises $N = 10$ simulations, each containing 20 random arrangements of particles and 2000 water molecules.

**MR imaging and relaxometry**. Imaging and relaxometry were performed on a Bruker 7 T MRI scanner. A 72 mm diameter volume coil was used to both transmit and receive RF signals. To measure the $T_2$ relaxation rate of RAW cells after nanoparticle labeling, the cells were labeled identically to their preparation for NV magnetometry, then trypsonized, resuspended in 10 mL DMEM and pelleted for 5 min at 350 g. DMEM was aspirated and cells were resuspended in 150 μL PBS. The cells were mixed with an equal number of unsupplemented cells during resuspension in PBS to extend the $T_2$ time of the final pellet, improving the fidelity of the $T_2$ fit. After transferring the cells to a 300 μL centrifuge tube, the cells were pelleted for 5 min at 350 × g. These tubes were embedded in a phantom comprising 1% agarose dissolved in PBS and imaged using a multi-echo spin-echo (CPMG) sequence (TR = 4000 ms, TE = 11 ms, 2 averages, 20 echoes, 273 × 273 × 1000 μm voxel size). $T_2$ relaxation was obtained from a monoexponential fit of the first 6 echoes. In order to establish the intrinsic $T_2$ of RAW cell pellets for our Monte Carlo simulations, we measured the $T_2$ relaxation of 4 pellets of unsupplemented RAW cells using the same parameters as above, except that, since the $T_2$ was significantly longer, we fitted the first 20 echoes. Fitting using only even echoes produced the same results as fitting all echoes (Supplementary Fig. 8).

For the scenario in which nanoparticles are unclustered in the extracellular space, unsupplemented RAW cells were pelleted and resuspended in PBS supplemented with 100 μg ml$^{-1}$ IONs. This concentration was selected to ensure a measurable $T_2$ and allow both in silico and in cellulo comparisons between the per-iron relaxation rates of extracellular and internalized particle scenarios. The validity of a per-iron comparison was confirmed by previous studies of the linearity of relaxivity for this size of iron oxide nanoparticles when unclustered[47]. To limit endocytosis, cells were moved to the cold MRI bore and imaged immediately after supplementation and pelleting. Imaging parameters were as described above.

**Mouse model of iron overload**. Animal experiments were conducted under a protocol approved by the Institutional Animal Care and Use Committee of the California Institute of Technology. Female C57bl/6 mice were injected in the tail vein with 10 mg kg$^{-1}$ of dragon green labeled 900 nm ION (Bangs) or saline. A total of three mice were used in this study. No randomization or blinding were needed given the design of the study. Eighteen hours after injection, the mice were perfused with 20 mL of 10% neutral buffered formalin, and their livers were collected for MRI or NV magnetometry. MRI was performed on livers embedded in 1% agarose using the 7T scanner described above, using a spin-echo pulse sequence with TR = 2500 ms, TE = 11 ms, 4 averages, and a 273 × 273 × 1000 μm voxel size. For NV magnetometry, the liver was frozen in OCT embedding media and sectioned into 10 μm slices. Sections were mounted in on glass coverslips. We inverted the glass cover slip and pressed the tissue sample against the NV diamond. Silicon vacuum grease was applied at the edge of the cover slip (away from the diamond) to hold the sample against the diamond. After this preparation was complete, PBS was added to the dish to wet the sample. We performed fluorescent imaging to locate magnetic sources in the tissue. As the sources were sparsely distributed, the camera was set to an autogain function to allow for short exposure time and rapid scanning. The camera was set back to fixed gain before NV imaging commenced. To compensate for magnetic field sources being further from the diamond due to tissue thickness and/or folds in the sections, NV imaging was performed with a strong (25 mT) bias field applied along a single NV axis. This strong bias field served to increase the magnetization of the magnetic inclusions in the liver. As it was applied along an NV axis, this bias field did not significantly reduce the contrast of the relevant ODMR spectral lines. However, such a strong bias field precludes the use of vector magnetometry. Future improvements to histological sample preparation should increase the sample flatness and bring the magnetic material closer to the diamond surface, allowing for a lower bias field and, as a result, vector magnetometry and source localization. Images were acquired with a Zeiss 40× near infrared water immersion objective (NA 0.8).

**Software and image processing**. All fits and plots were generated in MATLAB. Monte Carlo Simulations were performed in C++ on a Linux High Performance Computing Cluster.

**Statistical analysis**. Sample sizes were chosen on the basis of preliminary experiments to have sufficient replicates for statistical comparison. Data are plotted, and values are given in the text, as mean ± S.E.M. Statistical comparisons assumed similar variance.

**Code availability**. All the relevant software scripts are available from the authors upon request.

**Data availability**. All the relevant data are available from the authors upon request.

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

## Acknowledgements

We acknowledge Arnab Mukherjee, George Lu, Vivek Bharadwaj, My Linh Pham, Andrei Faraon, Geoffrey Blake, Joe Kirschvink, Manuel Monge, Hans Gruber, Michael Tyszka, Russ Jacobs, and John Wood for helpful discussions. This work was supported by the National Science Foundation Graduate Research Fellowship (P.R.), Caltech Center for Environmental–Microbial Interactions (M.G.S.), the Burroughs Wellcome Fund (M. G.S.), the NSF EPMD and PoLS programs (R.L.W.), and the U. S. Army Research Laboratory and the U. S. Army Research Office under contract/grant number W911NF1510548 (R.L.W.). Research in the Shapiro Laboratory is also supported by the Heritage Medical Research Institute, the Pew Scholarship in the Biomedical Sciences and the David and Lucile Packard Fellowship for Science and Engineering.

## Author contributions

H.C.D., P.R and M.G.S. conceived and planned the study with input from J.F.B., D.R.G. and R.L.W. H.C.D., P.R. and J.F.B. constructed the magneto-microscope. H.C.D. performed the NV magnetometry experiments and analyzed the resulting data. H.C.D., P.R. and A.L.-G. prepared the in vitro and in vivo specimens. P.R. and H.C.D. performed the MRI measurements and analyzed the resulting data. H.C.D. and A.B. developed and

performed the Monte Carlo simulations. H.C.D., P.R. and M.G.S. wrote the manuscript with input from all other authors.

## Additional information

**Competing interests:** The authors declare no competing financial interests.

