## [Peer Review File · Nature Communications]

Reviewers' comments:

Reviewer #1 (Remarks to the Author):

This manuscript addresses the root cause of MR relaxation of water protons due to spatially-varying magnetic fields. To date, the distribution of these fields from sources such as iron oxide nanoparticles has been difficult to empirically establish, leading to inferences drawn from modeling or from the macroscopic effects of these fields on the proton relaxation rate.

The authors employ a novel technique to map these magnetic fields using NV-center magnetometry to achieve sub-micron resolution. In addition, they localize magnetic field sources and use Monte Carlo simulation of the same to predict the macroscopic relaxation rates from their mapped fields. These techniques are then tested on two models: iron uptake by macrophages, and iron overload in the liver of a murine model.

It would be valuable to expand this line of research to look at other types of field sources that may not be as "well-behaved" as these particular IONs. For instance, these particles do not appear to be strongly interacting, are of fairly large size (200 nm diameter), and the behavior at clinical field strengths can be predicted by extrapolating measurements made at the DC bias field of 10 mT by use of the M vs H curve. However, in vivo iron may or may not exhibit similar behaviors, especially if not superparamagnetic.

Of critical importance is the distribution, geometry, and magnetic order of in vivo iron, such as ferritin and hemosiderin. Additionally, the use of iron-based contrast agents already on the market for clinical use (e.g., Feridex, Feraheme) or pre-clinical would be useful. Knowing how well the techniques described in the manuscript extend to these cases would be valuable. Smaller particles with different magnetic ordering might not generate sufficient magnetic field for NV-center detection (17 nT sensitivity reported here). This could be briefly visited in the discussion, or saved for later work.

The paper is well-written, with an appropriate distribution of information contained in the main text and in the supplement.

Line 208- please define ODMR as optically-detected magnetic resonance. I don't believe it is defined prior.

Line 286-287- The free water diffusivity at 12.9 °C can be predicted from the Speedy-Angell fit in the Holz paper. However, this value is likely higher than the extracellular diffusivity. The intracellular diffusivity was set to 1/3 of the extra cellular diffusivity (as currently listed) without justification. What effect might changing these have on the MC results? You might also mention that your MC simulation assumes a purely outer-sphere-like relaxation mechanism, i.e., there is no residence time on the surface of the IONs.

Reviewer #2 (Remarks to the Author):

The paper by Davis et al. uses NV magnetometry to image the local magnetic field within mammalian cells produced by known MRI contrast agents. The goal is to exploit the higher resolution of optical microscopy and the detailed knowledge of the magnetic field within the cell to more precisely predict the macroscale MRI contrast.

The overall science seems sound but I am a bit skeptical on the novelty and overall impact. The

application of NVs to determine the magnitude and direction of external magnetic fields has already been demonstrated extensively in a variety of settings (I am sure the authors know the literature in this area is vast). In particular, ensembles of near-surface NVs (identical to those used in this work) have been exploited to image cell with sub-cellular resolution, not only by the Walsworth group (listed as Refs 13 and 14) but also by others (e.g., Magnetic spin imaging under ambient conditions with sub-cellular resolution, Steinert et al., Nature Commun. 4, 1607 (2013)). The authors indicate that none of these experiments were carried out in mammalian cells and that no connection with MRI contrast was made before. However, since the presented results are also a proof of principle, I am not fully convinced this work represents a major step. In particular I don't see (in practice) a strong case on how to transition from the comparatively-lower-resolution of MRI images to NV magnetometry or vice versa (in Fig. 3 the authors ultimately have to excise a section of the liver). Something similar can be said about Fig. 4 where high-resolution microscopy is possible thanks to the NVs but no direct comparison can be made with MRI. Finally, at times I found the overall presentation unnecessarily obscure to the non-expert, particularly in the abstract and introductory section (e.g., statements such as "this technique can follow the magnetic consequences of a dynamic process in living cells" or others similarly confusing are not particularly helpful to guide the reader).

Reviewer #3 (Remarks to the Author):

The authors have developed methods to directly map subcellular magnetic fields in live and fixed mammalian cells and tissues, using nitrogen vacancy (NV) microscopy, and have then used these microscopic fields to accurately estimate macroscale T2 relaxation times.

The methodology is novel and would be of interest to others in the community. The ability to accurately estimate T2 relaxation times from microscopic magnetic field measurements would be of value in understanding MRI contrast as well as in the development of improved MR contrast agents.

The work is relatively convincing, although I have a number of questions and requests for clarification that I believe would improve this work and make it more convincing. I have also made comments in a few places that are intended to improve the ability for other researchers to understand and replicate the work.

Specific Comments

Page 3 line 35: "the resonance frequency of these transitions shifts linearly". Can the authors be more clear about how they are exploiting this Zeeman-based shift in resonance frequency?

Page 4 line 52: "and an in vivo model". This is an overstatement and I'd like the authors to change the wording so as not to mislead the readership. The authors have NOT demonstrated in vivo NV microscopy, and yet the wording in the paper could easily be misinterpreted this way. The model of iron load to the liver of mice is of course an in vivo model, but in my opinion not worth explicitly using this phrase "in vivo" here, since it is obvious that it is an in vivo model (as with any other mouse model is). All that the authors have demonstrated is the ability to map fields in excised histological tissue sections.

Page 4 line 64: "10mT" – is this field high enough? What other aspects of this bias field are important? Authors should comment on the optimal strength and characteristics (such as uniformity) of the bias field.

Page 4 line 66: "superparamagnetic IONs" – authors should describe these IONs in enough detail that

others could replicate their experiments.

Page 4 line 70: "field profiles ... cannot be directly observed by conventional EM". Granted that is true; however, EM or perhaps other forms of microscopy could be used to locate particles, even within cells, and then local fields could be accurately modeled around these particles based on the known physics of superparamagnetic particles in the presence of a bias field. T2 relaxation could then be predicted in the same way that the authors are doing here. So in other words, I think the authors may be overstating the importance of NV microscopy as a way of predicting T2 relaxation. I'd like the authors to discuss this, either here or in Discussion or both.

Page 4 line 72: "after incubating them ..." – authors should provide data about the loaded cells, e.g. the loading level in units of pgFe/cell, any other characteristics of potential relevance and interest to readers involved with cellular MRI.

Page 4 line 79: "3D maps of magnetic field sources in the sample" – these maps should correspond to maps of ION particle location / size, I believe, and could be validated using EM or other microscopy of the same samples / cells. I would like to see validation of this type. Can authors do this and comment on this?

Page 5 line 84: "Then, the off-diamond height (z) ..." – this seems like a potentially very ill-posed problem. Authors should show an analysis of this. For example, it sounds like it would be difficult to distinguish a different size particle from a different z-height. If the ION cores are not monodisperse (and they usually aren't) then this seems like it could be a significant problem to me.

Page 5 line 98: "Importantly, since this information ..." - But is it possible to make NV measurements and T2 measurements on the same sample? Also, how many cells (or what density of cells) can be measured with the NV method?

Page 5 line 101: What was a 1:1 mixture necessary and what would the consequences of using a different mixture be?

Page 5 line 104: "Monte Carlo prediction" - It is unlikely that cells are arranged in such a regular cartesian grid of equal-sized spheres. Authors should comment on the changes in T2 that arise from irregular distribution and size of cells.

Page 5 line 106: "concentration of IONs" - However, T2 could have been predicted from knowledge of particle location/size within a cell, which might for example be measurable by EM or other light microscopy. The same Monte Carlo prediction of T2 could be done from this knowledge, which would therefore not require NV microscopy. Authors need to justify why NV microscopy is necessary for understanding the basics of T2 in ION-loaded cells.

Page 5 line 110: Fig. 2f – the figure is labeled "clustered" but this is ambiguous. IONs could be clustered extracellularly or could be confined / clustered inside sub-cellular compartments. I believe these two situations should yield different T2 values but the authors should comment. Also, how did their simulations deal with the barriers / restrictions to water motion introduced by sub-cellular compartment membranes? One relevant simulation that I'd like to see would be to show simulated T2 values for extracellularly clustered Fe particles, and to compare to intracellularly compartmentalized Fe particles.

Page 5 line 118: "900nm" – why so large? Is NV measurement / microscopy not possible using more conventional IONs? This is not a particle size that is commonly used for in vivo "clinical" applications of

IONs.

Page 5 line 123: "punctate" – doesn't look punctate to me in Fig 3c. In fact the images in Fig 3c are quite poor – I don't recognize structures in the fluorescent images (they are very poor quality) and don't understand what correlations I'm supposed to be making between the magnetic field maps and those fluorescent images. Authors need to improve image quality and make those correlations more clear.

Page 5 line 138: "This allowed us ..." - Similar to a comment above, why is it not possible to perform source localization in this situation - that would be more informative and would allow a more direct validation against optical or EM, and more direct confirmation that "clustering" or sub-cellular compartmentalization is being measured.

Page 5 line 145: Discussion - This Discussion is very brief. I would like to see more, such as a more detailed discussion of limitations (such as the ill-posed nature of source localization and why source localization was not performed on the tissue samples), also comparison to any other technologies for microscopy of magnetic fields such as microSQUID, discussion of sensitivity, spatial resolving power for sources, and temporal resolution.

Page 8 line 174: "unknown" – why is this not known? Seems like it would be important to know this, to allow others to replicate this work. Authors should clarify.

Page 9 line 207: "sufficient out-of-plane field strength ..." – authors should be more clear about what this means - how much is "sufficient" and what are the specifications / tolerance limits for this bias field uniformity or other characteristics.

Page 9 line 208: "ODMR" – this acronym has not been defined.

Page 9 line 215: "quadratic background ..." – how is bias magnet inhomogeneity corrected for at the source localization stage?

Page 9 line 221: "sensitivity" – how was this sensitivity determined? It is good to see this sensitivity expressed in units of ION particles that can be detected at a given spacing off the diamond; however, commonly used IONs have iron oxide cores of 3-5nm, not 200nm. Authors should express sensitivity in units of these common IONs since that is where the general interest would lie. If this NV microscopy method is not going to be able to sense conventional 3-5nm ION cores, then this is a significant limitation that needs to be discussed.

Page 9 line 223: "200nm" – does this refer to the iron core size? If so, that seems too large compared to standard/practical IONs commonly used for cell tracking.

Page 9 line 227: "the bias field was aligned ..." – not clear why this was done, and why the same method as in the cellular imaging (with bias field parallel to plane of NV layer) could not be used.

Page 11 line 285: "12.9degC" – seems like a pretty cold temperature. Why was this particular temperature used?

Page 11 line 285: should also specify intracellular/extracellular volume fractions.

Page 11 line 288: should specify units of the bulk spin magnetization; the summation implies that this is unitless but does that make sense?

Page 12 line 294: "Data presented ..." – can the authors justify that these numbers in this sentence are large enough?

Page 12 line 308: "multi-echo spin echo" – was this a CPMG sequence? Authors should be more clear. Authors should comment on the fact that T2 measurements will depend on echo spacing in this kind of measurement.

Page 12 line 309: "T2 relaxation ..." – this seems to imply that the signal decay was not monoexponential. Authors should clarify. Also, the same may have been true for the simulated NMR signal decay and should be commented on also.

Page 12 line 312: "unsupplemented ..." – were these live or fixed cells? If live cells, wouldn't the IONs have been taken into the cells over time?

Page 13 line 328: "to seal ..." – does this imply that a vacuum was used to force the tissue against the diamond? If so, authors should be more clear.

Page 13 line 331: "single NV axis" – which axis specifically?

Page 13 line 332: "This strong ..." – why wouldn't this strong bias field have been used for all experiments, not just this one? Seems like it would be produce stronger fields and therefore better measurement precision.

Page 18 figure 3 caption: "a, h and i" – I don't see these labels.

Page 19 figure 4: Is it not possible to show optical (e.g. brightfield) images at all the same time points as the magnetic field images? Authors should specify how large scale bars are.

Page 19 figure 4 and associated text in the main body: Is it not possible to localize magnetization sources in this situation of tissue sample (like they did for cell samples?). If not, why not?

We thank the referees for their overall enthusiasm and constructive comments, which have helped improve this manuscript. Author responses are provided below in blue text. Revisions to the manuscript are also highlighted with blue font.

Reviewer #1 (Remarks to the Author):

This manuscript addresses the root cause of MR relaxation of water protons due to spatially-varying magnetic fields. To date, the distribution of these fields from sources such as iron oxide nanoparticles has been difficult to empirically establish, leading to inferences drawn from modeling or from the macroscopic effects of these fields on the proton relaxation rate.

The authors employ a novel technique to map these magnetic fields using NV-center magnetometry to achieve sub-micron resolution. In addition, they localize magnetic field sources and use Monte Carlo simulation of the same to predict the macroscopic relaxation rates from their mapped fields. These techniques are then tested on two models: iron uptake by macrophages, and iron overload in the liver of a murine model.

It would be valuable to expand this line of research to look at other types of field sources that may not be as "well-behaved" as these particular IONs. For instance, these particles do not appear to be strongly interacting, are of fairly large size (200 nm diameter), and the behavior at clinical field strengths can be predicted by extrapolating measurements made at the DC bias field of 10 mT by use of the M vs H curve. However, in vivo iron may or may not exhibit similar behaviors, especially if not superparamagnetic. Of critical importance is the distribution, geometry, and magnetic order of in vivo iron, such as ferritin and hemosiderin. Additionally, the use of iron-based contrast agents already on the market for clinical use (e.g., Feridex, Feraheme) or pre-clinical would be useful. Knowing how well the techniques described in the manuscript extend to these cases would be valuable. Smaller particles with different magnetic ordering might not generate sufficient magnetic field for NV-center detection (17 nT sensitivity reported here). This could be briefly visited in the discussion, or saved for later work.

Thank you for this comment. Since the focus of this manuscript was on developing the basic methodology for connecting microscale magnetic fields with MRI signals, we used macrophage uptake of well-behaved magnetic materials as our primary model to facilitate detailed experimental validation of this method. However, we agree that in future studies it will be important to extend the capabilities of this technique beyond this basic scenario to less ideal magnetic materials, including biogenic iron. As suggested, we have expanded our discussion of the need for these future experiments and the technology improvements needed to make such measurements possible. This text appears in the updated manuscript on lines 175 to 196.

The paper is well-written, with an appropriate distribution of information contained in the main text and in the supplement.

Thank you!

Line 208- please define ODMR as optically-detected magnetic resonance. I don't believe it is defined prior.

Thank you. The definition has been added on line 253 in the revised manuscript.

Line 286-287- The free water diffusivity at 12.9 °C can be predicted from the Speedy-Angell fit in the Holz paper. However, this value is likely higher than the extracellular diffusivity. The intracellular diffusivity was set to 1/3 of the extra cellular diffusivity (as currently listed) without justification. What effect might changing

these have on the MC results? You might also mention that your MC simulation assumes a purely outer-sphere-like relaxation mechanism, i.e., there is no residence time on the surface of the IONs.

Thank you for this comment. Our diffusion model is based on the work of Pfeuffer et al ¹, who found that extracellular water is well approximated by the bulk water self-diffusion coefficient, and that the intracellular self-diffusion constant is approximately 1/3 that of bulk water. The effect of cell membranes and the geometric arrangement of cells is explicitly included in our cellular lattice model, in which the cell membranes are treated as semi-permeable barriers to water. This modeling approach produced experimentally validated results in a recent study of cellular water diffusion for MRI reporter development². The relevant citations have been added to the text on lines 356-360.

We have also added a comment on the assumption of purely outer-sphere relaxation on lines 345-346.

Thank you for very much for your review.

Reviewer #2 (Remarks to the Author):

The paper by Davis et al. uses NV magnetometry to image the local magnetic field within mammalian cells produced by known MRI contrast agents. The goal is to exploit the higher resolution of optical microscopy and the detailed knowledge of the magnetic field within the cell to more precisely predict the macroscale MRI contrast.

The overall science seems sound but I am a bit skeptical on the novelty and overall impact. The application of NVs to determine the magnitude and direction of external magnetic fields has already been demonstrated extensively in a variety of settings (I am sure the authors know the literature in this area is vast). In particular, ensembles of near-surface NVs (identical to those used in this work) have been exploited to image cell with sub-cellular resolution, not only by the Walsworth group (listed as Refs 13 and 14) but also by others (e.g., Magnetic spin imaging under ambient conditions with sub-cellular resolution, Steinert et al., Nature Commun. 4, 1607 (2013)). The authors indicate that none of these experiments were carried out in mammalian cells and that no connection with MRI contrast was made before.

Thank you for your comments. We acknowledge that our study builds on previous work by multiple groups establishing the basic technology for wide-field mapping of magnetic fields near diamond surfaces, which we tried to convey appropriately in the introduction, including work done with mammalian cells (e.g. ref 14). We regret that we missed the Steinert citation due to our focus on DC magnetometry; we have added it in the revised manuscript on lines 45-48 and 197-203, and discuss it further below.

The primary novelty of our study is the development of methods to make the connection between NV-based maps of the microscale distribution of magnetic fields in biological samples with the contrast seen in conventional MRI. We believe this is an important new capability for the MRI field, and were happy to see this view supported by most of the referees.

However, since the presented results are also a proof of principle, I am not fully convinced this work represents a major step. In particular I don't see (in practice) a strong case on how to transition from the comparatively-lower-resolution of MRI images to NV magnetometry or vice versa (in Fig. 3 the authors ultimately have to excise a section of the liver). Something similar can be said about Fig. 4 where high-resolution microscopy is possible thanks to the NVs but no direct comparison can be made with MRI.

Thank you for this comment. The primary evidence that the methods presented in this study effectively connect lower-resolution MR images to NV magnetometry is provided in the detailed experiments presented in Figures 1 and 2, as well as Supplementary Figures 2, 3, 7 and 8. In these experiments, we focused on demonstrating the core capability of our method using the well-controlled experimental scenario of iron

oxide phagocytosis by macrophages. This allowed us to prepare biological samples with different experimentally controlled distributions of magnetic materials, enabling us to establish and validate the ability of NV magnetometry, 3-D magnetic field reconstruction, and Monte Carlo simulations of nuclear spin precession to accurately predict MRI contrast. This key demonstration is further reinforced in the revised manuscript with the addition of several experiments suggested by Reviewer #3 (Supplementary Figs 7 and 8).

The experiments shown in Figs. 3 and 4 are intended to demonstrate the potential for future applications in more complex or dynamic contexts. While we certainly agree that more work must be done in the future to truly apply this technology to answer specific biological and clinical questions, we believe the proof-of-concept nature of the experiments presented in this manuscript are consistent with what has been done in other high-impact publications in this field (e.g. ³⁻⁸).

We also thank the reviewer for pointing out the 2013 paper from Steinert et al, which used NV relaxation measurements to map the location of T₁ contrast agents near the diamond surface. We apologize for omitting this citation in our original manuscript, and have added a reference to it on lines 45-48. We have also added a discussion starting on line 197 about how this complementary method for evaluating T₁ contrast could be combined with the presented work on T₂ contrast to create a more complete picture of potential MRI contrast sources in biological specimens.

Finally, at times I found the overall presentation unnecessarily obscure to the non-expert, particularly in the abstract and introductory section (e.g., statements such as “this technique can follow the magnetic consequences of a dynamic process in living cells” or others similarly confusing are not particularly helpful to guide the reader).

Thank you for this comment. We have revised the manuscript as suggested to provide more precise information to readers (e.g., lines 13 to 14 and 54 to 57).

Thank you for your helpful comments.

Reviewer #3 (Remarks to the Author):

The authors have developed methods to directly map subcellular magnetic fields in live and fixed mammalian cells and tissues, using nitrogen vacancy (NV) microscopy, and have then used these microscopic fields to accurately estimate macroscale T₂ relaxation times.

The methodology is novel and would be of interest to others in the community. The ability to accurately estimate T₂ relaxation times from microscopic magnetic field measurements would be of value in understanding MRI contrast as well as in the development of improved MR contrast agents.

Thank you for these comments!

The work is relatively convincing, although I have a number of questions and requests for clarification that I believe would improve this work and make it more convincing. I have also made comments in a few places that are intended to improve the ability for other researchers to understand and replicate the work.

Specific Comments

Page 3 line 35: “the resonance frequency of these transitions shifts linearly”. Can the authors be more clear about how they are exploiting this Zeeman-based shift in resonance frequency?

Thank you. We have clarified this sentence to explain more precisely that the Zeeman energy difference between the +1 and -1 states leads to the splitting of the 2.87 GHz resonance into two peaks, whose

separation from each other increases linearly with magnetic field strength (lines 36 to 38).

Page 4 line 52: “and an in vivo model”. This is an overstatement and I’d like the authors to change the wording so as not to mislead the readership. The authors have NOT demonstrated in vivo NV microscopy, and yet the wording in the paper could easily be misinterpreted this way. The model of iron load to the liver of mice is of course an in vivo model, but in my opinion not worth explicitly using this phrase “in vivo” here, since it is obvious that it is an in vivo model (as with any other mouse model is). All that the authors have demonstrated is the ability to map fields in excised histological tissue sections.

Thank you. As suggested, we have modified the text to clarify that we used histological specimens from a mouse model of hepatic iron overload (lines 54-57 and abstract.).

Page 4 line 64: “10mT” – is this field high enough? What other aspects of this bias field are important? Authors should comment on the optimal strength and characteristics (such as uniformity) of the bias field.

Thank you for this question. To better explain our choice of bias field in the main text, we have added text on lines 68 to 70 and 416 to 419. To further expand on this topic, selecting a bias field for vector magnetometry of superparamagnetic sources requires balancing the need to magnetize the particles with the need to maintain NV fluorescence contrast, since the sensitivity of NV vector magnetometry decreases at higher field strengths. This effect is due to the quantum basis of the NV defect shifting from along the defect axis to along the applied field, so that m_s is no longer an eigenstate of the spin Hamiltonian at higher field strengths. The eigenstates of the spin Hamiltonian will be mixed states of $m_s=0$ and $m_s=\pm 1$, reducing the contrast of the optically detected electronic spin resonance. This was discussed at length in Tetienne et al (reference 17 in the manuscript) and is summarized in our revised SI section on *SQUID Magnetometry and Saturation Field Scaling*.

Page 4 line 66: “superparamagnetic IONs” – authors should describe these IONs in enough detail that others could replicate their experiments.

Thank you. We have clarified in the main text that these are multi-core superparamagnetic 200 nm iron oxide particles (lines 78-79), and added further information in the Methods, including the catalog number of the particles (line 241). In addition, we have characterized the magnetic behavior of these particles using SQUID magnetometry, as shown in Supplementary Fig. 2.

Page 4 line 70: “field profiles ... cannot be directly observed by conventional EM”. Granted that is true; however, EM or perhaps other forms of microscopy could be used to locate particles, even within cells, and then local fields could be accurately modeled around these particles based on the known physics of superparamagnetic particles in the presence of a bias field. T2 relaxation could then be predicted in the same way that the authors are doing here. So in other words, I think the authors may be overstating the importance of NV microscopy as a way of predicting T2 relaxation. I’d like the authors to discuss this, either here or in Discussion or both.

Thank you for this suggestion. We have added a discussion of this topic on lines 171-183, emphasizing that while this study used particles with known properties in experimentally controlled scenarios to validate our method, the technique we have developed can be applied to magnetic field sources with unknown properties, which would not be possible with electron microscopy or other localization methods that do not directly measure magnetic fields.

Page 4 line 72: “after incubating them ...” – authors should provide data about the loaded cells, e.g. the loading level in units of pgFe/cell, any other characteristics of potential relevance and interest to readers involved with cellular MRI.

Thank you. As suggested, we have added data on the iron loading level in (pg/cell), reported on lines 98-101 and discussed in the methods on lines 285-295. We note that these results match up very well with the iron concentration predicted from NV magnetometry, further validating the accuracy of our NV data fitting procedures.

Page 4 line 79: “3D maps of magnetic field sources in the sample” – these maps should correspond to maps of ION particle location / size, I believe, and could be validated using EM or other microscopyp of the same samples / cells. I would like to see validation of this type. Can authors do this and comment on this?

Thank you for this suggestion. In order to provide independent confirmation of particle localization, we fluorescently labeled the nanoparticles and performed an additional set of vector magnetometry experiments together with direct fluorescent detection of the particles. These results are provided in Supplementary Fig. 8 and referenced in the main text on lines 97 to 98. In brief, we found a strong correspondence between the (x,y) localization of our fitting technique and the centroid of the fluorescent signal from the magnetic clusters (with an average offset of 790 ± 105 nm). We believe this modest offset is due to physical deformation of the diamond and attached glass due to heating by the laser during NV measurements, and future work improving drift correction could help improve the fidelity between fluorescence imaging and NV magnetic field source localization. 3D fluorescence localization is not possible in our NV microscopy setup because it is not confocal, and moving the samples to a confocal microscope while keeping the precise field of view was not feasible. Likewise, it was not feasible to keep spatial registration with thin-sectioned EM samples. However, in addition to this 2-D localization data, the accuracy of our NV data fits is corroborated by independent measurements of average iron content, as discussed in our response to the previous question.

Page 5 line 84: “Then, the off-diamond height (z) ...” – this seems like a potentially very ill-posed problem. Authors should show an analysis of this. For example, it sounds like it would be difficult to distinguish a different size particle from a different z-height. If the ION cores are not monodisperse (and they usually aren't) then this seems like it could be a significant problem to me.

Thank you for this comment. Although this was not immediately intuitive to us when we started the project, the problem of z-localization is, in-fact well-posed in the vast majority of scenarios. As clarified in the revised text on lines 316 to 319, and demonstrated with a proof in the Supplement (equations S5 to S10), the fact that the magnetic field is measured at multiple points in the x-y plane of the diamond, and that the impact on this planar field pattern is affected differently by height and magnetization, allows both the height off the diamond and strength of the field source to be estimated. As shown in the proof in SI, there does not exist a pair of different (M,z) values such that the magnetic field is the same for (M, z) and (M',z') for all (x,y) coordinates in the measurement plane. A basic assumption of this model is that the sources can be modeled as point-like. For scenarios including sparse particle labeling and nanoparticle clusters internalized by cells, this is justified. However, for more diffuse deposits of magnetic material, a more refined fitting approach may be required. The need for future work in this direction is stated in our revised Discussion on lines 181-183.

Page 5 line 98: “Importantly, since this information ...” - But is it possible to make NV measurements and T2 measurements on the same sample? Also, how many cells (or what density of cells) can be measured with the NV method?

Thank you for this question. Measuring the same exact sample with both NV magnetometry and MRI would be very difficult with existing apparatus. Even if one could develop a method to section a biological specimen corresponding to a single MRI voxel, the size of a typical high resolution voxel has dimensions of $(100 \mu\text{m})^3$, while the typical field of view and of NV magnetometry represents a volume of $\sim (20 \mu\text{m} \times 20 \mu\text{m} \times 10 \mu\text{m})$. For this reason, we believe the methodology presented in this study, which allows a biological specimen to be sampled using NV magnetometry and used as input for predictive modeling of transverse relaxivity, is

important and advantageous for the task at hand. However, as it is now becoming possible using AC NV magnetometry to measure nuclear spin precession, it is conceivable that future studies could measure both the magnetic field and water relaxation in the same specimen. This possibility is discussed in the revised manuscript on lines 201 to 203.

Page 5 line 101: What was a 1:1 mixture necessary and what would the consequences of using a different mixture be?

We mixed the supplemented cells with unsupplemented cells to maintain a measurable T2 in our MRI scanner. Unmixed cells could be used if loaded with other magnetic sources. Other mixtures would be expected to have predictable effects on T2. We have added a comment to this effect on lines 113 to 114 of the main text and lines 380 to 382 of the methods section.

Page 5 line 104: "Monte Carlo prediction" - It is unlikely that cells are arranged in such a regular cartesian grid of equal-sized spheres. Authors should comment on the changes in T2 that arise from irregular distribution and size of cells.

Thank you. We agree that this is a significant simplification, which has nevertheless produced useful predictions in other studies modeling MRI contrast. We have added a comment on lines 339 to 341, drawing attention to the simplification and citing other papers in which this or similar simplifications have been used.

Page 5 line 106: "concentration of IONs" - However, T2 could have been predicted from knowledge of particle location/size within a cell, which might for example be measurable by EM or other light microscopy. The same Monte Carlo prediction of T2 could be done from this knowledge, which would therefore not require NV microscopy. Authors need to justify why NV microscopy is necessary for understanding the basics of T2 in ION-loaded cells.

Thank you. Please see our response to the question regarding page 4 line 79, above, addressing this issue.

Page 5 line 110: Fig. 2f – the figure is labeled "clustered" but this is ambiguous. IONs could be clustered extracellularly or could be confined / clustered inside sub-cellular compartments. I believe these two situations should yield different T2 values but the authors should comment. Also, how did their simulations deal with the barriers / restrictions to water motion introduced by sub-cellular compartment membranes? One relevant simulation that I'd like to see would be to show simulated T2 values for extracellularly clustered Fe particles, and to compare to intracellularly compartmentalized Fe particles.

Thank you for this insight. To address this question, we have added 3 new simulations to the manuscript. These simulations are described in detail in the SI, with their results potted in Supplementary Figure 7. They are also referenced in the main text on lines 123 to 125.

Page 5 line 118: "900nm" – why so large? Is NV measurement / microscopy not possible using more conventional IONs? This is not a particle size that is commonly used for in vivo "clinical" applications of IONs.

Thank you for this question. As we have clarified in the revised text on lines 131 to 132, our goal in administering these particles was to create a proof-of-concept scenario for punctate iron oxide deposits in the liver rather than a clinical scenario involving ION administration for contrast imaging. Using 900 nm particles allowed us to obtain such punctate deposition while producing reasonable T2 contrast in MRI. In addition, the detection of larger particles with NV magnetometry is more robust in our proof-of-concept experiments to issues such as imperfect flattening of histological samples against the diamond. The optimization of histological preparations for future studies of smaller and biogenic field sources in tissues is

discussed on lines 191 to 196. Additionally, we cite several major studies in which micron-sized iron oxide particles were used as contrast sources in in vivo MRI experiments (line 190).

Page 5 line 123: “punctate” – doesn’t look punctate to me in Fig 3c. In fact the images in Fig 3c are quite poor – I don’t recognize structures in the fluorescent images (they are very poor quality) and don’t understand what correlations I’m supposed to be making between the magnetic field maps and those fluorescent images. Authors need to improve image quality and make those correlations more clear.

We have clarified in the figure legend and accompanying text (lines 138-139) that the punctate features of the iron-loaded liver images shown in the top part of Fig. 3c (and in the additional tissue sections in Supplementary Fig. 4) are the individual magnetic dipoles (strong local minima and maxima next to each other) observed in the NV image and the single fluorescent source observed in the fluorescence image. By comparison, no punctate features are observed in the non-iron loaded control tissue. Note that the non-uniform background in the NV image reflects the underlying noise floor of the NV diamond measurement, likely due to higher-order strain effects. Signal from histological samples is weaker than in cultured cells because magnetic sources are offset further from the diamond surface. The saline magnetometry image shows no significant magnetic field signal, showing only the background noise in the diamond. The top fluorescent image in Fig. 3c shows a bright spot corresponding to the location of the nanoparticle cluster, while the lower fluorescent image just shows the background tissue autofluorescence (which appears brighter than in the top image because fluorescent images were acquired with automatic gain). Further discussion of the fluorescent imaging of the tissue sections has been added to the figure caption and to the methods section (lines 409 to 412),

Page 5 line 138: “This allowed us ...” - Similar to a comment above, why is it not possible to perform source localization in this situation - that would be more informative and would allow a more direct validation against optical or EM, and more direct confirmation that "clustering" or sub-cellular compartmentalization is being measured.

Thank you for this comment. This limitation stem from the fact that keeping cells alive requires limiting the light dose applied to the specimen, resulting in a tradeoff between SNR, temporal resolution and the number of NV axes along which the magnetic field is measured. For this proof-of-concept experiment in live cells, we chose to increase temporal resolution while imaging the magnetic field along a single NV axis, which allows dynamic imaging of cluster formation and movement within the cells, but not localization in 3-D. We have added text on lines 153 to 156 to make this limitation explicit. We believe live-cell vector magnetometry is achievable with increased SNR, which is within reach for future studies with improvements described in the discussion on lines 191-196.

Page 5 line 145: Discussion - This Discussion is very brief. I would like to see more, such as a more detailed discussion of limitations (such as the ill-posed nature of source localization and why source localization was not performed on the tissue samples), also comparison to any other technologies for microscopy of magnetic fields such as microSQUID, discussion of sensitivity, spatial resolving power for sources, and temporal resolution.

Thank you for this suggestion. We have significantly expanded the discussion, lines 167-203, in response to this comment and other questions and suggestions from the referee.

Page 8 line 174: “unknown” – why is this not known? Seems like it would be important to know this, to allow others to replicate this work. Authors should clarify.

We have added this information to the text on lines 218 to 220 after getting it from the diamond manufacturer.

Page 9 line 207: “sufficient out-of-plane field strength ...” – authors should be more clear about what this means - how much is "sufficient" and what are the specifications / tolerance limits for this bias field uniformity or other characteristics.

Thank you. We have augmented our explanation of this step in the methods section, lines 252-256, to specify why an out-of-plane field component was necessary. We have also clarified on lines 262 to 264 how any non-uniformity in the bias field is subtracted from magnetometry results, and included an estimate of our background bias field gradient (based on a representative image).

Page 9 line 208: “ODMR” – this acronym has not been defined.

Thank you for pointing this out. The definition has been added on line 253 of the revised manuscript.

Page 9 line 215: “quadratic background ...” – how is bias magnet inhomogeneity corrected for at the source localization stage?

We have revised the text on lines 261-264 to clarify that inhomogeneities in the bias field are accounted for prior to the source localization stage by subtracting out linear and quadratic gradients from the image, which works well because the bias field has a much lower spatial frequency than expected signals.

Page 9 line 221: “sensitivity” – how was this sensitivity determined? It is good to see this sensitivity expressed in units of ION particles that can be detected at a given spacing off the diamond; however, commonly used IONs have iron oxide cores of 3-5nm, not 200nm. Authors should express sensitivity in units of these common IONs since that is where the general interest would lie. If this NV microscopy method is not going to be able to sense conventional 3-5nm ION cores, then this is a significant limitation that needs to be discussed.

Sensitivity was determined by measuring the standard deviation of sequential images. As the magnetic field at a point scales linearly with the magnetic moment of the source and the magnetic moment scales with the volume of the particle, the minimum particle size that our current apparatus could measure with confidence 10 μm off of the diamond surface would be 92 nm. At the diamond surface, this detection limit comes down to 10 nm and is largely limited by the thickness of the NV layer. We have extended our discussion of sensitivity on lines 184 to 196.

Page 9 line 223: “200nm” – does this refer to the iron core size? If so, that seems too large compared to standard/practical IONs commonly used for cell tracking.

This does refer to the total iron core size. (These particles are multi-core, with the individual core size not provided by the manufacturer). The nanoparticle size was chosen to provide good SNR from single nanoparticles anywhere inside of the cell, while being within the range of particles used in MRI cell labeling. While USPIOs are widely used, other studies have relied on detection of hundred-nm and micron sized IONS. We have discussed this point and added references ⁹, ¹⁰, and ¹¹ to the text on lines 74-76 and 189-191.

Page 9 line 227: “the bias field was aligned ...” – not clear why this was done, and why the same method as in the cellular imaging (with bias field parallel to plane of NV layer) could not be used.

Thank you. This is discussed in our response to the question above concerning Page 5, line 138.

Page 11 line 285: “12.9degC” – seems like a pretty cold temperature. Why was this particular temperature used?

We have clarified on line 358-359 that this is the temperature in our magnet bore, and was chosen to maintain a correspondence between modeling and experiments.

Page 11 line 285: should also specify intracellular/extracellular volume fractions.

Thank you for pointing out this oversight. We have added the intracellular volume fraction on line 339.

Page 11 line 288: should specify units of the bulk spin magnetization; the summation implies that this is unitless but does that make sense?

Bulk Spin Magnetization is normalized to the magnetic moment of a single water molecule for arithmetic convenience. We have added this to the text for clarity on line 359-560.

Page 12 line 294: "Data presented ..." – can the authors justify that these numbers in this sentence are large enough?

Thank you for this question. We selected these number so that the uncertainty in the mean value was lower than the uncertainty in the mean of the experimental trials to which these simulations were compared. We have clarified this choice in the text on lines 368-369.

Page 12 line 308: "multi-echo spin echo" – was this a CPMG sequence? Authors should be more clear. Authors should comment on the fact that T2 measurements will depend on echo spacing in this kind of measurement.

Yes this is a CPMG sequence. We have revised the text for clarity on line 377. The combined effects of echo spacing and spatial frequency of the T2 contrast agent are indeed highly relevant, and we have pointed this out on lines 347 to 348.

Page 12 line 309: "T2 relaxation ..." – this seems to imply that the signal decay was not monoexponential. Authors should clarify. Also, the same may have been true for the simulated NMR signal decay and should be commented on also.

Thank you. Both our experiments and simulations exhibited monoexponential decay starting with the first echo in the CPMG sequence. As observed in previous studies, before the first echo a rapid unrefocusable dephasing of spins in very close proximity to magnetic field sources causes additional relaxation that is not part of the monoexponential fit.

Page 12 line 312: "unsupplemented ..." – were these live or fixed cells? If live cells, wouldn't the IONs have been taken into the cells over time?

The cells were live but were imaged immediately after supplementation and were transported to the MRI on ice to prevent endocytosis. It is also unlikely that the cells would actively endocytose during imaging given the temperature in the magnet bore. This has been clarified in the revised text on lines 394-395.

Page 13 line 328: "to seal ..." – does this imply that a vacuum was used to force the tissue against the diamond? If so, authors should be more clear.

Vacuum grease was used on the edge of the cover slip to hold the sample still and tight to the diamond. No vacuum was used. We have adjusted the text for clarity on lines 407-408.

Page 13 line 331: "single NV axis" – which axis specifically?

The NV defect exists in 4 distinct orientations in a tetrahedral configuration. We probed the axis aligned with the applied bias field. We selected the bias field orientation based on convenience in our apparatus, but this choice is arbitrary.

Page 13 line 332: “This strong ...” – why wouldn't this strong bias field have been used for all experiments, not just this one? Seems like it would be produce stronger fields and therefore better measurement precision.

Thank you for this question. Our histological measurements were performed with the bias field aligned along a single NV axis, and we collected data from only that axis. This allowed us to boost the field strength without producing off-axis fields deleterious to NV fluorescence contrast. We could not use the same approach for vector magnetometry, where bias field had on-axis and off-axis components for all four NV axes and could therefore not be too strong, as discussed in response to an earlier question.

Page 18 figure 3 caption: “a, h and i” – I don't see these labels.

Thank you for drawing this to our attention. The figure has been corrected.

Page 19 figure 4: Is it not possible to show optical (e.g. brightfield) images at all the same time points as the magnetic field images? Authors should specify how large scale bars are.

We have added the bright field images from the long time-course study to the supplement. As the brightfield lamp was adjusted before magnetometry commenced, there is a change in illumination for the later images. The “fast” study did not allow sufficient time in-between scans for bright field imaging with our current setup. Thank you for pointing out the omission of the scale bar size. We have added this to the figure caption.

Page 19 figure 4 and associated text in the main body: Is it not possible to localize magnetization sources in this situation of tissue sample (like they did for cell samples?). If not, why not?

This question is addressed in our response to the question above concerning Page 5, line 138.

Thank you very much for your detailed and helpful feedback.

Citations:

1. Pfeuffer, J., Flögel, U., Dreher, W. & Leibfritz, D. Restricted diffusion and exchange of intracellular water: theoretical modelling and diffusion time dependence of ¹H NMR measurements on perfused glial cells. *NMR in Biomedicine* **11**, 19-31 (1998).
2. Mukherjee, A., Wu, D., Davis, H.C. & Shapiro, M.G. Non-invasive imaging using reporter genes altering cellular water permeability. *Nature Communications* **7**, 13891 (2016).
3. Le Sage, D. et al. Optical magnetic imaging of living cells. *Nature* **496**, 486--489 (2013).
4. Glenn, D.R. et al. Single-cell magnetic imaging using a quantum diamond microscope. *Nat Meth* **12**, 736-738 (2015).
5. Steinert, S. et al. Magnetic spin imaging under ambient conditions with sub-cellular resolution. **4**, 1607 (2013).
6. Hall, L.T. et al. Monitoring ion-channel function in real time through quantum decoherence. *Proceedings of the National Academy of Sciences of the United States of America* **107**, 18777-18782 (2010).
7. I. Lovchinsky, A.O.S., 1, 2* E. Urbach, 1 N. P. de Leon, 1, 2† S. Choi, 1 K. De Greve, 1 R. Evans, 1 R. Gertner, 2 E. Bersin, 1 C. Müller, 3 L. McGuinness, 3 F. Jelezko, 3 R.. L. Walsworth, 1,4,5 H. Park, 1,2,5, 6‡ M. D. Lukin1‡ Nuclear magnetic resonance detection and spectroscopy of single proteins using quantum logic. *Science* **351**, 836-842 (2016).
8. Kucsko, G. et al. Nanometre-scale thermometry in a living cell. *Nature* **500**, 54-58 (2013).
9. McAteer, M.A. et al. In vivo magnetic resonance imaging of acute brain inflammation using microparticles of iron oxide. *Nature Medicine* **13**, 1253-1258 (2007).
10. Shapiro, E.M. et al. MRI detection of single particles for cellular imaging. *Proceedings of the National Academy of Sciences of the United States of America* **101**, 10901-10906 (2004).
11. Tarulli, E. et al. Effectiveness of micron-sized superparamagnetic iron oxide particles as markers for detection of migration of bone marrow-derived mesenchymal stromal cells in a stroke model. *Journal of Magnetic Resonance Imaging* **37**, 1409-1418 (2013).

Reviewers' comments:

Reviewer #1 (Remarks to the Author):

The authors have sufficiently addressed my original comments. Having read the responses to the other reviewers, I would like to point out that the typical high resolution for MRI cited to reviewer 3 of $100 \times 100 \times 100 \mu\text{m}$ is true for clinical MRI, but micro-MRI, especially as performed in a vertical bore magnet, may be capable of significantly higher resolution, on the order of the NV magnetometry resolution.

With regards to the use of CPMG: it is important to only use even echoes. There is mention of the "first echo" in the response to reviewer comments.

Line 132: "efficient" iron loading, or "sufficient" iron loading.

Line 345: make explicit mention that inner-sphere effects can lead to additional dephasing that will not be reversible by use of typical CPMG sequences (as they occur on a time-scale much faster than 180° refocusing pulses period)

Line 347-348: This sentence sounds awkward and poorly justified. The interecho spacing may indeed affect T2 measurements in the case of chemical exchange (i.e., Carver-Richards model) or very large, localized field gradients. It is not obvious to me that this was tested to see if it applies to the case at hand, as only one τ_{cp} was used for modeling (5.5 ms). There is no citation to justify the "...affect the efficacy of the refocusing pulses..." sentence. Refocusing pulses don't care about τ_{cp} , they are as efficient as they will be at $100 \mu\text{s}$ as they are at 100ms . What matter is how well you can sample the T2 decay. This sentence should be revised to read something along the lines of "Adjusting the Carr-Purcell time can affect the determination of T2".

Reviewer #3 (Remarks to the Author):

The revisions adequately address my original comments.

We appreciate the reviewers' helpful suggestions. Responses are provided below in blue text. Updated sections of the manuscript are also highlighted in blue font.

Reviewer #1:

The authors have sufficiently addressed my original comments. Having read the responses to the other reviewers, I would like to point out that the typical high resolution for MRI cited to reviewer 3 of 100x100x100 μm is true for clinical MRI, but micro-MRI, especially as performed in a vertical bore magnet, may be capable of significantly higher resolution, on the order of the NV magnetometry resolution.

We agree that micro-MRI is capable of higher spatial resolution than clinical MRI. However, it is still difficult to get into the single-micron regime in liquid samples, where resolution is limited to $\sim 6 \mu\text{m}^1$. More typically, biological sample imaging, even *ex vivo*, has larger than $\sim 50 \mu\text{m}$ resolution². We have added a statement about this in the discussion on lines 167–169.

With regards to the use of CPMG: it is important to only use even echoes. There is mention of the "first echo" in the response to reviewer comments.

We agree that fitting only even echoes can improve T2 determination in typical imaging practice by reducing any errors introduced by the imperfect amplitude of hard pi pulses³. However, this does not appear to be an issue in our experiments, where we had ample opportunity for pulse calibration. To confirm that this is the case, we re-fitted representative samples of ION-supplemented and unsupplemented cells using only even echoes, and compared the fitted T2 values to the those resulting from fitting both odd and even echoes. This new data is provided in Supplementary Figure 9. No significant difference was found between the fitted T2 values. This data is referenced in the revised manuscript on line 386.

Line 132: "efficient" iron loading, or "sufficient" iron loading.

We have removed this superfluous clause to avoid confusion.

Line 345: make explicit mention that inner-sphere effects can lead to additional dephasing that will not be reversible by use of typical CPMG sequences (as they occur on a time-scale much faster than 180° refocusing pulses period)

Additional language has been added, as suggested, making explicit that water molecules susceptible to the inner sphere effect are rapidly dephased and cannot be refocused using the CPMG sequence. (Lines 342-344).

Line 347-348: This sentence sounds awkward and poorly justified. The interecho spacing may indeed affect T2 measurements in the case of chemical exchange (i.e., Carver-Richards model) or very large, localized field gradients. It is not obvious to me that this was tested to see if it applies to the case at hand, as only one tau_cp was used for modeling (5.5 ms). There is no citation to justify the "...affect the efficacy of the refocusing pulses..." sentence. Refocusing pulses don't care about tau_cp, they are as efficient as they will be at 100 μs as they are at 100 ms. What matter is how well you can sample the T2 decay. This sentence should be revised to read something along the lines of "Adjusting the Carr-Purcell time can affect the determination of T2".

We have modified the sentence as suggested.

Reviewer #3:

The revisions adequately address my original comments.

1. Moore, E. & Tycko, R. Micron-scale magnetic resonance imaging of both liquids and solids. *Journal of magnetic resonance (San Diego, Calif. : 1997)* **260**, 1-9 (2015).
2. Deans, A.E., Wadghiri, Y.Z., Aristizábal, O. & Turnbull, D.H. 3D mapping of neuronal migration in the embryonic mouse brain with magnetic resonance microimaging. *NeuroImage* **114**, 303-310 (2015).
3. Bain, A.D., Kumar Anand, C. & Nie, Z. Exact solution of the CPMG pulse sequence with phase variation down the echo train: Application to R2 measurements. *Journal of Magnetic Resonance* **209**, 183-194 (2011).

REVIEWERS' COMMENTS:

Reviewer #1 (Remarks to the Author):

I have read the responses, the related main body text, and examined suppl. Figure 9. I am satisfied.

Two things to the authors for *future* consideration (does not need to be changed in this article):

1. Sup. Fig. 9d clearly shows that the even echoes more closely lie to the fitted curve. A plot of the residuals of 9c and 9d would make this apparent (for the sake of curiosity). This is indicative of imperfect pi pulses (they are almost always imperfect). Future work should employ proper CPMG technique and exclude odd echoes (which is the point of the Meiboom-Gill addition to the Carr-Purcell approach).

2. I recommend plotting semilog plots of the T2 decay: deviations from the exponential decay will be more readily apparent with the "straight-line" decay a semilog plot (on the vertical axis) yields